# Two-Step Preparation of CCF/PEEK Wrapped Yarn for 3D Printing Composites with Enhanced Mechanical Properties

**DOI:** 10.3390/ma16031168

**Published:** 2023-01-30

**Authors:** Jianghu Zhang, Hao Shen, Lili Yang, Dengteng Ge

**Affiliations:** 1State Key Laboratory for Modification of Chemical Fibers and Polymer Materials, College of Materials Science and Engineering, Donghua University, Shanghai 201620, China; 2Institute of Functional Materials, Donghua University, Shanghai 201620, China

**Keywords:** 3D printing, CCF/PEEK, wrapping yarn, double spinning

## Abstract

Continuous fiber reinforced thermoplastic composites (CFTPCs) have shown advantages such as high strength, long life, corrosion resistance, and green recyclability. Three-dimensional printing of CFTPCs opened up a new strategy for the fabrication of composites with complicated structures, low cost, and short production cycles. However, a traditional 3D printing process usually causes poor impregnation of the fiber or surface damage of the fiber due to the short impregnation time or high viscosity of the thermoplastic resin. Here, continuous carbon fiber/poly(ether-ether-ketones) (CCF/PEEK) wrapped yarn was fabricated via powder impregnation and using double spinning technology for the 3D printing. The concentration of PEEK powder suspension and wire speed were optimized as 15% and 2.0 m/min. The twist of wrapped yarn was optimized as 1037 T/m. Mechanical testing showed that the 3D-printed composite wire had excellent tensile and bending strength, which was about 1.6~4.2 times larger than those without the powder pre-impregnation process. It is mainly attributed to the improved impregnation of the CF which took place during the powder pre-impregnation process. We believe that our research on wrapped yarn for 3D-printed composites provides an effective strategy for the 3D printing of composites with enhanced mechanical properties.

## 1. Introduction

Continuous fiber reinforced thermoplastic composites (CFTPCs) have received tremendous attention due to the advantages of having high strength, long life, corrosion resistance, and green recyclability [1,2,3,4,5]. CFTPCs have been widely used in aerospace, transportation, and high precision processing equipment. In particular, as representatives of high-performance fiber and semi-crystalline thermoplastic resin, continuous carbon fiber reinforced poly(ether-ether-ketone) composites (CCF/PEEK) play an important role because of their excellent mechanical properties, thermal stability, and chemical resistance [6,7]. Several methods have been developed for the manufacturing of CFTPCs, such as resin transfer molding, injection molding, winding molding, pultrusion forming, etc. However, these traditional manufacturing processes have the defects of complicated processing, long production cycles, and high cost. In contrast, 3D printing is an advanced additive manufacturing technology that enables the rapid manufacture of complex structural parts, making it possible to achieve the integrated manufacturing of high-performance composite materials [8,9,10,11,12,13,14,15]. According to the pretreatment method of continuous fibers, there are two basic strategies for the 3D printing of CFTPCs: in situ impregnation and pre-impregnation methods. In the former approach, the wetting of thermoplastic resin and fibers is poor due to the high viscosity of polymer and the short impregnation time during 3D printing. In the matter process, however, the yarn is rigid after the pre-impregnation, causing surface damage of the fiber and mechanical property degradation.

To address these problems, some apparatuses in a 3D printing system such as the wire feed printhead have been improved. Moreover, the printing parameters have been further optimized. In addition, some supporting processes, i.e., ultrasonic vibration, laser-assisted bonding [16,17,18], and hot isostatic pressing, have been introduced to decrease the porosity, and enhance the mechanical properties. These passive strategies improved the properties to some extent, but the higher cost, more complex process, and moderate properties are the main shortcomings. In this research, a novel active approach is proposed using CCF/PEEK wrapped yarn to enhance the wetting and keep the flexibility of the impregnated fiber. Here, CCF/PEEK wrapped yarn was prepared via a two-step method including wet powder impregnation and the use of double spinning technology. One-step wrapping technologies such as CCF/PEEK [19] and CGF/PEEK [20,21] have been studied; however, the wetting of fiber and resin or the performance of composites was moderate [22,23,24,25,26]. Furthermore, the influence of preparation parameters on the mechanical properties of final composites has not yet been reported in detail. In order to further improve resin impregnation and enhance the mechanical properties of composites, a two-step method including powder impregnation and a double spinning process was carried out for the wrapping of the yarn. The key parameters in these processes were optimized. After a surface treatment of the wrapped yarn, composite wires were prepared via 3D printing. The results show that the power impregnation process plays a more important role in the uniform wetting of resin on fibers, and the mechanical preparties of 3D-printed wires are 1.6~4.2 times of those without powder pre-impregnation process.

## 2. Materials and Methods

### 2.1. Materials

Carbon fiber (CF, T300B-1K) was purchased from Toray Co., Ltd. (Tokyo, Japan). PEEK powders (150 P) were obtained from Victrex Co., Ltd. (Lancashire, UK). PEEK fiber (23tex/30F) was purchased from Changzhou Co-win New Material Technology Co., Ltd. (Changzhou, China). Acetone (99.7%), N. N-dimethylformamide (99.7%) and polyvinyl alcohol was obtained from Sinopharm Chemical Reagent Co., Ltd. (Shanghai, China).

### 2.2. Two-Step Fabrication of CCF/PEEK Wrapped Yarn

The preparation of CCF/PEEK wrapped yarn included two steps: powder impregnation and a subsequent double spinning process. The PEEK powder suspension was achieved via the mixture of PEEK powder, polyvinyl alcohol, and deionized water. The mass fraction of polyvinyl alcohol was 2%, and the mass fraction of PEEK powder was set to 5%, 10%, 15%, 20% and 25%, respectively. Then, the CCF was passed through the powder suspension at a speed of 0.5 m/min, 1.0 m/min, 1.5 m/min, 2.0 m/min and 2.5 m/min, respectively, followed by a drying process. The CCF/PEEK wrapped yarn was prepared via a double spinning process on a covering yarn machine produced by Shaoxing Hengrun Textile Machinery Factory. The powder impregnated CF was used as the core yarn and PEEK fiber as the outer wrapping yarn. Different twists of wrapped yarn were obtained due to the modulation of the wire speed.

### 2.3. Surface Treatment of CCF/PEEK Wrapped Yarn

The CCF/PEEK wrapped yarn was firstly desized via immersion into acetone at 85 °C for 12 h. The sizing solution was obtained from the dissolution of poly (aryl ether ketone) (PAEK) powder in N, N-dimethylformamide with a 1% mass fraction. Then, the desized wrapped yarn was soaked into sizing agent solution for 30 min at room temperature. After a drying process, the CCF/PEEK wrapped yarn could be used for 3D printing.

### 2.4. Three-Dimensional Printing of Composite Wire

CCF/PEEK composite wire was prepared through a 3D printing nozzle (ENDER-3, Shenzhen Creality Technology Co., Ltd. in Shenzhen, China). The nozzle diameter was 0.4 mm, the nozzle temperature was 380 °C, and the extrusion speed was set to 0.1, 0.2, 0.3, 0.4, and 0.5 m/min, respectively.

### 2.5. Characterization

The morphologies were observed with an optical microscope (TH4-200, Olympus, Tokyo, Japan) and scanning electron microscope (S4800, Hitachi, Tokyo, Japan). The distance between the upper and lower clamps were 250 mm and the tensile speed was 250 mm/min. The friction performance of the fiber was tested using a friction performance tester (Model 339, Shenzhen Speedre Technology Co., Ltd. in Shenzhen, China). A 45 g roller and 800 mesh sandpaper were used. The reciprocating speed was 60 times/min and the friction length was 55 mm. The surface elements were analyzed using an X-ray photoelectron spectrometer (ESCALAB 250Xi, Thermo Fisher Scientific, Waltham, MA, USA). The fiber surface wettability was measured using a contact angle measuring instrument (SL200KB, USA KINO Industry Co., Ltd. in Boston, MA, USA). The mechanical properties of the composite wire were tested using a universal testing machine (RWT10, Shenzhen Reger Instrument Co., Ltd. in Shenzhen, China). In the tensile test, the distance between the upper and lower clamps was 150 mm, and the tensile speed was 1 mm/min. In the bending test, the test span was 6 mm, and the indenter pressing rate was 2 mm/min.

## 3. Results and Discussion

### 3.1. Preparation of CCF/PEEK Wrapped Yarn for 3D Printing

As illustrated in Figure 1, the preparation of CCF/PEEK wrapped yarn included two steps: powder impregnation of the CF and a double spinning process. After the surface sizing of the CCF/PEEK wrapped yarn, the CCF/PEEK composite wire was achieved via 3D printing. The main parameters in each step were optimized to achieve the best mechanical properties of the CCF/PEEK composites. In the first powder impregnation process, the mass fraction of the PEEK powder in the suspension and the wire moving speed greatly affect the impregnation effect [27]. Thus, the influence of the mass fraction of the PEEK powder in the suspension and the wire moving speed of the CF during impregnation were studied. The mass fraction of the CF was tested which indicates the ratio of ratio of the CF’s weight before the powder impregnation to the CF’s weight after the powder impregnation. As shown in Figure 2a, with the increase in the concentration of PEEK powder in the suspension, more PEEK particles are attached to CF, leading to a decrease in mass fraction of the CF. Too high or too low a CF mass fraction results in poor interface bonding or reduced mechanical properties. At a wiring speed of 2 m/min, here, the mass fraction of the PEEK powder of 15% was chosen, yielding a moderate CF mass fraction of 71%. In addition, the influence of the wire moving speed on the CF mass fraction was also studied, as shown in Figure 2b. With the increase in the wire speed, the mass fraction of the CF gradually increases due to the reduced adsorption time. However, in general, the CF mass fraction in the impregnated yarn is maintained in the range of 60–75%. When the speed exceeds 2 m/min, the increase in the CF mass fraction also tends to be gentle. Thus, a wire speed of 2.0 m/min was finally chosen. The SEM image of the CF after powder impregnation is shown in Figure 2c. It can be seen that a large number of PEEK particles are uniformly adsorbed both on the surface and the inside of the CF bundle. This is important for the mechanical properties of the final composites due to the better impregnation of fiber.

Double spinning technology is used here for the wrapping of the PEEK fiber around the CF surface. In this process, the powder-impregnated CF acts as the core yarn while the PEEK fiber is considered to be the outer wrapping yarn. As illustrated in Appendix A, the core yarn was moving at a constant rate, while a drawing roller of PEEK fiber rotated clockwise, and another drawing roller of PEEK fiber rotated counterclockwise. Thus the PEEK yarns were wrapped around the CF yarn to obtain the CCF/PEEK wrapped yarn. Figure 3a exhibits the image of the CCF/PEEK wrapped yarn. It can be seen that the external PEEK fiber is divided into two layers, which are helically wound outside the CF in the opposite direction. The twist of the CCF/PEEK wrapped yarn is determined by the winding rate of the PEEK yarn or the wire moving rate of the CF. Appendix A shows the different CCF/PEEK wrapped yarns obtained via the modulation of winding rate of the PEEK yarn. It is obvious that with the increase in the winding rate of the PEEK yarn, the surface of the CF is gradually covered by PEEK yarn.

The effects of wrapping yarn with different twists were studied. A tensile test was carried out on the pristine CF and CCF/PEEK wrapped yarn with a twist of 830 T/m. Figure 3b shows the relationships between the tensile force and strain. The tensile curve of the wrapped yarn is different from that of the CF, and the breaking of the wrapped yarn is mainly divided into two stages. In the first stage, the strength gradually increases with the increase in displacement until the CF as the core yarn breaks. Then, the strength decreases rapidly. In the second stage, the strength will increase again due to the support of external PEEK yarn wrapping. Compared with the CF, the breaking strength of the CF after wrapping is significantly increased. This is due to the low cohesion and friction force of the monofilament in CF during the stretching process, resulting in the monofilaments breaking separately. However, after wrapping, the PEEK fiber will form a certain pressure on the CF, which will increase the cohesion and friction between the monofilaments in CF, and finally form a large resultant force, which will greatly increase the breaking strength [28]. However, after wrapping, the external PEEK will form a certain pressure on the internal CF, so that the monofilaments in the CF bundle will be tightly held together. The friction inside the bundle will also increase, eventually enhancing the breaking strength. Figure 3c shows the CF mass fraction in wrapped yarns with different twists. With the increase in twist, more PEEK fiber is wrapped, and the CF mass fraction decreases.

Based on the friction and tensile tests, the friction numbers and breaking force of the CCF/PEEK wrapped yarn with different twists are summarized in Figure 3d,e. As shown in Figure 3d, the increase of twist of the CCF/PEEK wrapped yarn improves the friction resistance of wrapped yarn. This is because, during friction, the external fiber needs to be ground off before the internal CF can continue to be damaged. With the increase in twist, the content of the external PEEK fiber also gradually increases, and more friction time is needed to break the external PEEK fiber. In addition, when CF and wrapped yarn are rubbed for the same amount of time, most of the monofilaments in the CF will break, making CF invalid. However, only the PEEK fiber in wrapped yarn is damaged, and the CF in wrapped yarn remains intact. It can be demonstrated that the PEEK wrapping yarn not only increase the friction number, but also protects the internal carbon fiber from damaged in the subsequent processing. In Figure 3e, with the increase in wrapping twist, the breaking force of wrapped yarn increases first and then decreases slightly. When the twist is small, with the increase in twist, the axial inclination of the PEEK fiber to the CF increases, and the pressure from inward tightening increases, leading to the improvement in cohesion and friction inside the yarn. When the twist exceeds a certain value, the pre-stress of the fibers in the wrapped yarn would increase, and the mass fraction of the CF yarn would reduce, thus decreasing the breaking strength of the yarn.

In order to study the effect of surface treatment, the morphologies of wrapped yarn and pristine CF were firstly observed. As shown in Figure 4a, the original CF surface has a thin layer of the sizing agent, and its surface is relatively smooth. After desizing of the CF, there are a lot of grooves on the surface because the original sizing agent on the surface has been completely removed (Figure 4b). After sizing of the CF, a layer of the sizing agent appears on the CF surface, and the grooves on the fiber surface become blurred, which proves that the sizing agent has been successfully applied on the CF surface (Figure 4c). After the wrapped yarn is desized, the surface morphology of the CF inside the wrapped yarn is similar to that of pristine desized CF (Appendix A). It is proven that the CF has been successfully desized despite the wrapping of the PEEK yarn. In addition, the PEEK particles adsorbed on the CF in wrapped yarn will not be lost during desizing. This may be due to the good binding of covered PEEK yarn. At the same time, the external PEEK fiber also blocks the falling of PEEK powder. When the wrapped yarn is sized, a layer of sizing agent also appears on the surface of the internal CF, which indicates that the sizing agent can enter the CF through the gap in the external wrapped yarn. At the same time, the external PEEK fiber surface will also be covered with some sizing agent.

In order to further investigate the sizing process, an assessment of the surface chemical state and wettability of the CF wrapped yarn was carried out. Figure 5a–c show the C1S spectra of different CF surfaces. There are C-C, C-O, and C=O bonds on the surface of desized CF. After sizing, characteristic peaks representing phenyl (~284.9 eV), C-O-C (~286 eV), and O-C=O (~288.5 eV) appeared on the CF surface. This proves that the sizing agent (PAEK solution) was successfully applied to the CF surface. After sizing the CCF/PEEK wrapped yarn, these characteristic peaks also appear on the CF surface of the wrapped yarn. This proves that sizing agent can also be applied to CF in wrapped yarn. However, compared with the sized CF, the characteristic peak area of the CF in the sized wrapped yarn is smaller, which may be due to the blocking of the external PEEK fiber in wrapped yarn. The contact angle and surface energy of the CF surface are shown in Figure 5d,e. Among these, γ, γ^d^, and γ^p^ represent the surface energy, the dispersion component, and the polar component of surface energy, respectively. The contact angle of water and glycol on the desized CF is the highest, reaching 87°and 84°, respectively, and the surface energy is the lowest, at only 21.5 mN/m. This is because the CF surface after desizing is very rough, so the contact angle to the liquid is high and the surface energy is low [29]. In this case, the resin cannot easily flow and spread on the CF surface, which is not good for the mechanical properties of the composite. After sizing, the contact angle of the CF to the two liquids decreases, and the surface energy increases. In this case, the CF surface has good wettability, which is advantageous to the flow of resin on the CF surface. Similar results have been demonstrated that desizing leads to an increase in contact angle and a decrease in the surface energy, while after sizing, the contact angle decreases and the surface energy increases [29].

After sizing, the surface energy of the CCF/PEEK wrapped yarn is also higher than that of the desized carbon fiber, which proves that the sizing of the CCF/PEEK wrapped yarn was successful. In addition, compared with the direct sizing of CF, the surface energy of the wrapped yarn after sizing is lower. It is mainly due to the smaller amount of sizing agent coated on the CF surface in wrapped yarn. Furthermore, during the surface treatment of CCF/PEEK wrapped yarn, the PEEK fiber can protect the CF from damage.

### 3.2. Mechanical Performance of 3D Printing CCF/PEEK Composite Wire

CCF/PEEK composite wires were prepared using melt extrusion of the CCF/PEEK wrapped yarn through a 3D printing nozzle. The nozzle temperature was set to 380 °C. The TG curve of PEEK fiber is shown in Appendix A. It is deduced that the starting degradation temperature of PEEK is about ~410 °C. Thus, PEEK melts under a nozzle temperature of 380 °C but does not degrade. In order to prove the effect of the two-step method on the mechanical properties, composite yarns using the one-step method were also compared. In the one-step method, CF was directly wrapped with PEEK fiber without a powder impregnation process. Moreover, the effect of the pulling speed on the mechanical properties of composite wires were also studied. As shown in Figure 6a, the tensile strength of the composite wire via a two-step process is 2~3.6 times that of composite wire via a one-step process at the same pulling speed. While the bending strength of composite wire via a two-step process is 1.8~4.2 times that of composite wire via a one-step process at the same pulling speed. The mechanical properties of the composite wire decrease with the increase in pulling speed. When the pulling speed increases, the impregnation time of fiber and the melted PEEK resin decreases. Due to the high viscosity of PEEK resin, the resin cannot fully impregnate the CF under the high pulling speed, resulting in the reduction in the mechanical properties. In addition, the two-step method has better mechanical properties, compared with the CCF/nylon composite wire made by others [30]. Furthermore, compared with the wrapped yarn obtained via the one-step process, the mechanical properties of wrapped yarn made by the two-step method are more sensitive to the pulling speed. When the pulling speed increases, the mechanical properties decline more obviously. The cross-sectional morphologies of different composite wires are shown in Figure 6c–f. For the wrapped yarn via the one-step wrapping method, it remains difficult to permeate into the internal CF surface even with the increase in melt extrusion time. Only the external PEEK yarn melts, and very little PEEK permeates into the internal area, leading to a poor impregnation. Some pores could be even observed due to the non-infiltration of resin (Figure 6c,d). In contrast, for the wrapped yarn via the two-step method, the increase in melt extrusion time results in the better impregnation of the fiber due to heat transfer. Even at a high pulling speed, some fibers are still well impregnated and wrapped (Figure 6e). However, at a low pulling speed, the PEEK powder inside can be fully heated and melted, making the composite wire have a good impregnation effect (Figure 6f). It is suggested that the PEEK powder on the internal CF surface plays a major role in the impregnation process.

## 4. Conclusions

In this study, CCF/PEEK wrapped yarn was fabricated via a two-step method including powder impregnation and double spinning. CCF/PEEK composite wires were achieved via 3D printing after the surface treatment of the CCF/PEEK wrapped yarn. The key parameters of these processes were optimized for enhanced mechanical properties. The concentration of PEEK powder suspension and wire speed in the powder impregnation step were optimized as 15% and 2.0 m/min. The twist of wrapped yarn in the double spinning step was optimized as 1037 T/m. The mechanical testing of composite wires proves that powder impregnation step plays a more important role in the uniform wetting of resin on the CF surface, and the 3D-printed composite wire has a 1.6~4.2 times larger tensile and bending strength than those without a powder pre-impregnation process. These results highlight that our two-step method of wrapping yarn provides an effective strategy for the enhanced 3D printing of composites with excellent impregnation and flexibility of fibers. Moreover, several problems remain to be solved in future studies. For example, volatile surfactant could be replaced to be removed from the CF in the powder impregnation process. Better bonding between wrapped yarn should be studied in order to fabricate complicated composites. The flexibility of CCF/PEEK wrapped yarn will cause wire feeding difficulty. It is necessary to upgrade existing 3D printers to be more suitable for the printing of wrapped yarn.

## Figures and Tables

**Figure 1 materials-16-01168-f001:**
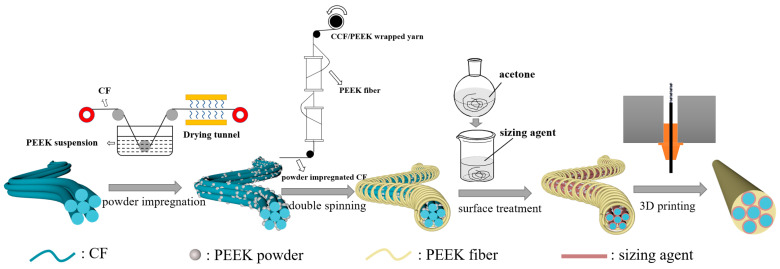
Schematic illustrations for the preparation of CCF/PEEK wrapped yarn and 3D-printed composite wire.

**Figure 2 materials-16-01168-f002:**
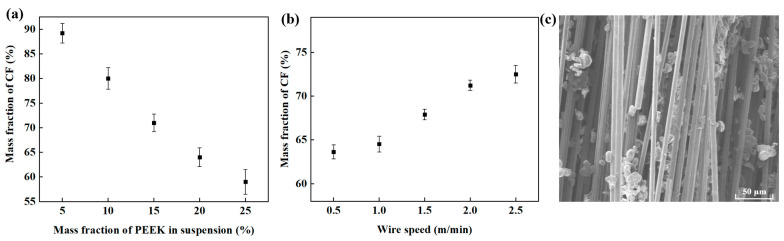
Effects of (**a**) PEEK powder mass fraction and (**b**) wire moving speed on the CF mass fraction in the powder impregnation process. (**c**) SEM image of CF after powder impregnation.

**Figure 3 materials-16-01168-f003:**
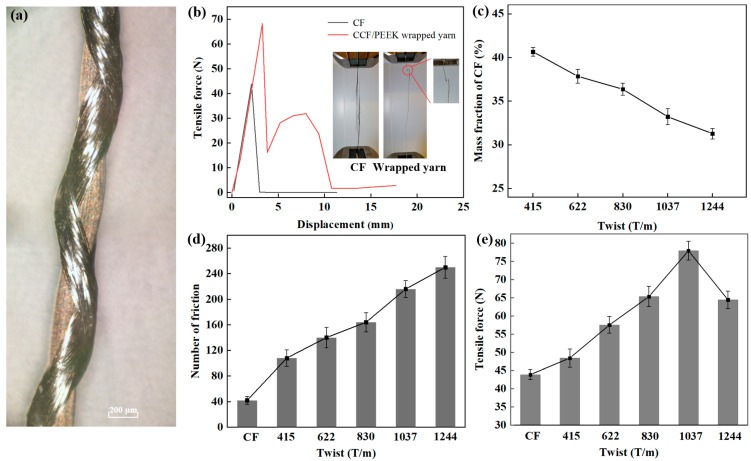
(**a**) Morphology of CCF/PEEK wrapped yarn. (**b**) Tensile curves of CCF/PEEK wrapped yarn and CF. Inset: pictures of tensile tests. (**c**) Mass fraction of CF in wrapped yarn with different twists. (**d**) Friction properties of CF and wrapped yarn with different twists. (**e**) Tensile strength of wrapped yarn with different twists.

**Figure 4 materials-16-01168-f004:**
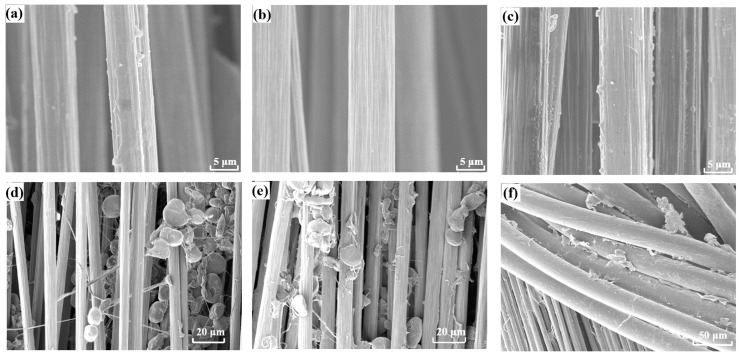
SEM images of (**a**) original, (**b**) desized, and (**c**) sized CF. (**d**–**f**) SEM images of (**d**) powder impregnated CCF/PEEK wrapped yarn, (**e**) after desizing, and (**f**) after sizing.

**Figure 5 materials-16-01168-f005:**
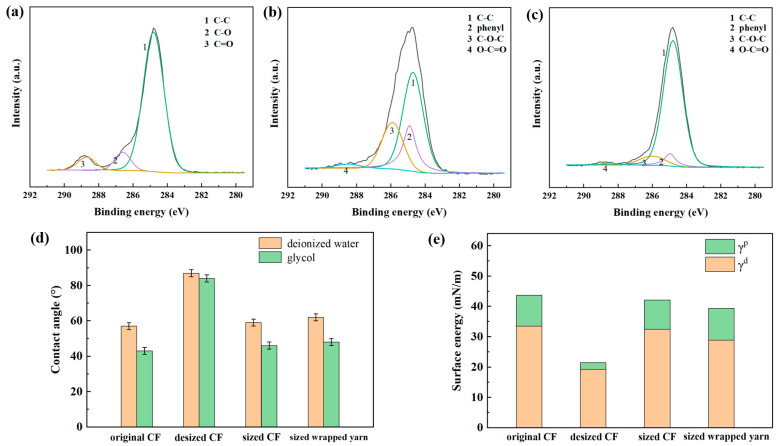
XPS spectra of (**a**) desized CF, (**b**) sized CF, and (**c**) sized CCF/PEEK wrapped yarn. (**d**) Contact angles and (**e**) surface energies of different fibers.

**Figure 6 materials-16-01168-f006:**
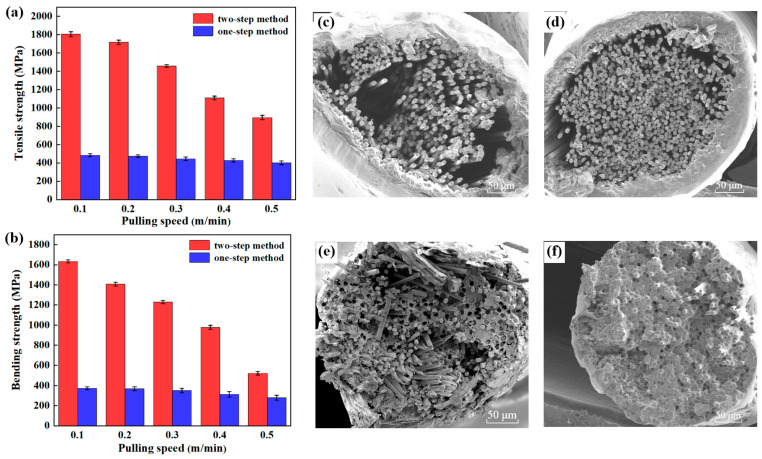
(**a**) Tensile strength and (**b**) bending strength of composite wires from different methods at different pulling speeds. (**c**–**d**) Cross-sectional SEM images of composite wire via one-step method at a pulling speed of (**c**) 0.5 m/min and (**d**) 0.1 m/min. (**e**–**f**) Cross-sectional SEM images of composite wire via two-step method at a pulling speed of (**e**) 0.5 m/min and (**f**) 0.1 m/min.

## Data Availability

Not applicable.

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
