# Peer review of "Two-Step Preparation of CCF/PEEK Wrapped Yarn for 3D Printing Composites with Enhanced Mechanical Properties"

_materials, 2023, doi:10.3390/ma16031168_

Round 1

Reviewer 1 Report

1. What's the temperature of yarn soaking in Section 2.3?

2. Please provide the 3D printer information in Section 2.4.

3. Any statistical analysis? How many samples were tested/characterized? How about the repeatability of results?

4. The font in the insets of Fig. 1 was too small to read. Please make it larger and consistent with other fonts in the figure. 

5. The authors used both "CF" and "CCF" in the manuscript. Any difference? If not, please make them consistent. 

6. How did the author know PEEK particles attached to CF that resulted in the decrease of mass fraction of CF? Was this an assumption? Please cite more papers to support.

7. Experimental details, such as the nozzle temperature, needed to be moved to Chapter 2. 

8. All references were cited in Introduction. The authors were recommended to cite more recently published papers in Results and discussion to support the phenomena, explanations, and statements.

Author Response

Item to item responses to reviewers’ comments:

Reviewer: 1

  1. What's the temperature of yarn soaking in Section 2.3?

Response: The wrapped yarn was soaked in sizing agent solution at room temperature. We added this point in section 2.3.

  1. Please provide the 3D printer information in Section 2.4.

Response: We have added the information of 3D printer in section 2.4.

“CCF/PEEK composite wire was prepared through a 3D printing nozzle (ENDER-3, Shenzhen Creality Technology Co., Ltd).”

  1. Any statistical analysis? How many samples were tested/characterized? How about the repeatability of results?

Response: During the experiments of mass fraction or mechanical properties, five samples were tested in each test. And the error bars were shown in Figure 2, 3 and 6.

  1. The font in the insets of Fig. 1 was too small to read. Please make it larger and consistent with other fonts in the figure. 

Response: This is a good suggestion. We have updated Fig. 1.

  1. The authors used both "CF" and "CCF" in the manuscript. Any difference? If not, please make them consistent. 

Response: Yeah, there is difference between CF and CCF. As pointed in the first sentence of maintext, CCF means “continuous carbon fiber” while CF just means “carbon fiber”. There are two kinds of fiber reinforced composites: continuous fiber reinforced composites and short fiber reinforced composites. Thus we pointed out continuous carbon fiber (CCF) to be different from short carbon fiber. In our manuscript, when we just mention the fiber, we marked as carbon fiber (CF) simply.

  1. How did the author know PEEK particles attached to CF that resulted in the decrease of mass fraction of CF? Was this an assumption? Please cite more papers to support.

Response: The method to characterize the mass fraction of CF is very simple. The mass faction of CF is the ratio of CF’s weight before the powder impregnation to CF’s weight after the powder impregnation. Due to the adsorption of PEEK particles, the mass fraction of CF decreases. In order to understand better, we have added some information in line 123~125.   

  1. Experimental details, such as the nozzle temperature, needed to be moved to Chapter 2. 

Response: Yeah, the details have been revised in section 2.

  1. All references were cited in Introduction. The authors were recommended to cite more recently published papers in Results and discussion to support the phenomena, explanations, and statements.

Response: We strongly agree with you. We have cited some related reports and discussed the results. For example, we cited Ref. [27] to discuss the results of powder impregnation process. We cited Ref. [28] to discuss the mechanical properties of wrapped yarn. We cited Ref. [29] to discuss the effect of sizing process on the surface state of fiber. We cited Ref. [30] to discuss the mechanical properties of 3D printed wire.

[27] Ho, K. K. C.; Shamsuddin, S. R.; Riaz, S.; Lamorinere, S.; Tran, M. Q.; Javaid, A.; Bismarck, A., Wet impregnation as route to unidirectional carbon fibre reinforced thermoplastic composites manufacturing. Plast Rubber Compos 2011, 40, 100-107. https://doi.org/10.1179/174328911X12988622801098.

[28] Lou, C. W.; Hu, J. J.; Lu, P. C.; Lin, J. H., Effect of twist coefficient and thermal treatment temperature on elasticity and tensile strength of wrapped yarns. Text Res J 2016, 86, 24-33. https://doi.org/10.1177/0040517514557315.

[29] Hassan, E. A. M.; Elagib, T. H. H.; Memon, H.; Yu, M.; Zhu, S., Surface Modification of Carbon Fibers by Grafting PEEK-NH2 for Improving Interfacial Adhesion with Polyetheretherketone. Materials 2019, 12, 778. https://doi.org/10.3390/ma12050778.

[30] Tian, X.; Zhang, Y.; Liu, T.; Li, D., Prepreg Preparation and 3D Printing of Continuous Carbon Fiber Reinforced Nylon Composite. Aeronaut Manuf Tech 2021, 64, 24-33. https://doi.org/10.16080/j.issn1671-833x.2021.15.024.

Reviewer 2 Report

There is no discussion in the manuscript. Results should be compared with the findings of the previously published articles 

Author Response

Item to item responses to reviewers’ comments:

Reviewer 2:

There is no discussion in the manuscript. Results should be compared with the findings of the previously published articles.

Response: We strongly agree with you. We have cited some related reports and discussed the results. For example, we cited Ref. [27] to discuss the results of powder impregnation process. We cited Ref. [28] to discuss the mechanical properties of wrapped yarn. We cited Ref. [29] to discuss the effect of sizing process on the surface state of fiber. We cited Ref. [30] to discuss the mechanical properties of 3D printed wire.

[27] Ho, K. K. C.; Shamsuddin, S. R.; Riaz, S.; Lamorinere, S.; Tran, M. Q.; Javaid, A.; Bismarck, A., Wet impregnation as route to unidirectional carbon fibre reinforced thermoplastic composites manufacturing. Plast Rubber Compos 2011, 40, 100-107. https://doi.org/10.1179/174328911X12988622801098.

[28] Lou, C. W.; Hu, J. J.; Lu, P. C.; Lin, J. H., Effect of twist coefficient and thermal treatment temperature on elasticity and tensile strength of wrapped yarns. Text Res J 2016, 86, 24-33. https://doi.org/10.1177/0040517514557315.

[29] Hassan, E. A. M.; Elagib, T. H. H.; Memon, H.; Yu, M.; Zhu, S., Surface Modification of Carbon Fibers by Grafting PEEK-NH2 for Improving Interfacial Adhesion with Polyetheretherketone. Materials 2019, 12, 778. https://doi.org/10.3390/ma12050778.

[30] Tian, X.; Zhang, Y.; Liu, T.; Li, D., Prepreg Preparation and 3D Printing of Continuous Carbon Fiber Reinforced Nylon Composite. Aeronaut Manuf Tech 2021, 64, 24-33. https://doi.org/10.16080/j.issn1671-833x.2021.15.024.

Moreover, we have added some suggestions for the future study as follows:

“Moreover, several problems remain to be solved in future study. For example, volatile surfactant could be replaced to be removed from the CF in the powder impregnation process. Better bonding between wrapped yarn should be studied in order to fabricate complicated composites. The flexibility of CCF/PEEK wrapped yarn will cause the wire feeding difficulty. It is necessary to upgrade the existing 3D printer to be more suitable for printing of wrapped yarn.”

Reviewer 3 Report

The authors studied the two-step preparation of CCF/PEEK-wrapped yarn for 3D printing composites with enhanced mechanical properties. The manuscript had an interesting topic and was well-written, however it could only be accepted with the following minor revisions:

1.    The introduction is very well; however, it doesn't include any information about additive manufacturing and FDM. Therefore, it is suggested the authors can add these papers as references; Investigation of ABS–oil palm fiber (Elaeis guineensis) composites filament as feedstock for fused deposition modeling, Rapid Prototyping Journal, 2020, (ahead-of-print); Effect of HBN fillers on rheology property and surface microstructure of ABS extrudate, Jurnal Teknologi, 84(4), 175-182.

2.    The novelty of the research was not highlighted in the text (introduction section). Please improve.

3.    Section 2.4, 3D printing of composite wire is referred. A 3D printer was used to fabricate CCF/PEEK, so what is the type of 3D printer / model / manufacturer?

4.    Section 3.1 “Preparation of CCF/PEEK wrapped yarn for 3D printing” and Figure 1 should be placed under Section 2 (materials and methods).

5.    Figure 6 c, d, e, f show SEM images of composite wire via one-step method, how about SEM images of wire made by two steps? They should be included in the paper to see the differences.

6.    Figure 5 d and e should be enlarged.

7.    Result and Discussion Section was very poor in citing the previous studies to compare with the present study (must be improved).

8.    The conclusion could be improved by adding the limitation, future study and implications for researchers.

Author Response

Item to item responses to reviewers’ comments:

Reviewer 3:

The authors studied the two-step preparation of CCF/PEEK-wrapped yarn for 3D printing composites with enhanced mechanical properties. The manuscript had an interesting topic and was well-written, however it could only be accepted with the following minor revisions:

  1. The introduction is very well; however, it doesn't include any information about additive manufacturing and FDM. Therefore, it is suggested the authors can add these papers as references; Investigation of ABS–oil palm fiber (Elaeis guineensis) composites filament as feedstock for fused deposition modeling, Rapid Prototyping Journal, 2020, (ahead-of-print); Effect of HBN fillers on rheology property and surface microstructure of ABS extrudate, Jurnal Teknologi, 84(4), 175-182.

Response: We have added these references as Ref. [14] and [15].

[14]. Ahmad, M. N.; Ishak, M. R.; Taha, M. M.; Mustapha, F.; Leman, Z., Investigation of ABS-oil palm fiber (Elaeis guineensis) composites filament as feedstock for fused deposition modeling. Rapid Prototyping J 2022. https://doi.org/10.1108/RPJ-05-2022-0164.

[15]. Lau, K. T.; Taha, M. M.; Abd Rashid, N. H.; Manogaran, D.; Ahmad, M. N., Effect of HBN fillers on rheology property and surface microstructure of ABS extrudate. Jurnal Teknologi 2022, 84, 175-182.

  1. The novelty of the research was not highlighted in the text (introduction section). Please improve.

Response: We have revised the Introduction section to highlight our novelty:

“In this research, a novel active approach is proposed based on CCF/PEEK wrapped yarn to enhance the wetting and keep the flexibility of impregnated fiber. Here, CCF/PEEK wrapped yarn were prepared via a two-step method including wet powder impregnation and following double spinning technology. The one-step wrapping technology such as CCF/PEEK [19], CGF/PEEK [20, 21] have been studied, however, the wetting of fiber and resin or the performance of composites were moderate [22-26]. Furthermore, the influence of preparation parameters on the mechanical properties of final composites has not yet been reported in details.”

  1. Section 2.4, 3D printing of composite wire is referred. A 3D printer was used to fabricate CCF/PEEK, so what is the type of 3D printer / model / manufacturer?

Response: We have added the detail of 3D printer (ENDER-3, Shenzhen Creality Technology Co., Ltd) in 2.4 section.

  1. Section 3.1 “Preparation of CCF/PEEK wrapped yarn for 3D printing” and Figure 1 should be placed under Section 2 (materials and methods).

Response: In section 3.1, we discussed the main parameters in each step including powder impregnation, double spinning, and surface treatment. We optimized the process parameters. Thus we can’t move the section 3.1 to section 2.

  1. Figure 6 c, d, e, f show SEM images of composite wire via one-step method, how about SEM images of wire made by two steps? They should be included in the paper to see the differences.

Response: The difference between samples via one-step method and via two-step method has been added in maintext.

  1. Figure 5 d and e should be enlarged.

Response: We have updated Figure 5d and 5e.

  1. Result and Discussion Section was very poor in citing the previous studies to compare with the present study (must be improved).

Response: We strongly agree with you. We have cited some related reports and discussed the results. For example, we cited Ref. [27] to discuss the results of powder impregnation process. We cited Ref. [28] to discuss the mechanical properties of wrapped yarn. We cited Ref. [29] to discuss the effect of sizing process on the surface state of fiber. We cited Ref. [30] to discuss the mechanical properties of 3D printed wire.

[27] Ho, K. K. C.; Shamsuddin, S. R.; Riaz, S.; Lamorinere, S.; Tran, M. Q.; Javaid, A.; Bismarck, A., Wet impregnation as route to unidirectional carbon fibre reinforced thermoplastic composites manufacturing. Plast Rubber Compos 2011, 40, 100-107. https://doi.org/10.1179/174328911X12988622801098.

[28] Lou, C. W.; Hu, J. J.; Lu, P. C.; Lin, J. H., Effect of twist coefficient and thermal treatment temperature on elasticity and tensile strength of wrapped yarns. Text Res J 2016, 86, 24-33. https://doi.org/10.1177/0040517514557315.

[29] Hassan, E. A. M.; Elagib, T. H. H.; Memon, H.; Yu, M.; Zhu, S., Surface Modification of Carbon Fibers by Grafting PEEK-NH2 for Improving Interfacial Adhesion with Polyetheretherketone. Materials 2019, 12, 778. https://doi.org/10.3390/ma12050778.

[30] Tian, X.; Zhang, Y.; Liu, T.; Li, D., Prepreg Preparation and 3D Printing of Continuous Carbon Fiber Reinforced Nylon Composite. Aeronaut Manuf Tech 2021, 64, 24-33. https://doi.org/10.16080/j.issn1671-833x.2021.15.024.

  1. The conclusion could be improved by adding the limitation, future study and implications for researchers.

Response: We have added some suggestions for the future study as follows:

“Moreover, several problems remain to be solved in future study. For example, volatile surfactant could be replaced to be removed from the CF in the powder impregnation process. Better bonding between wrapped yarn should be studied in order to fabricate complicated composites. The flexibility of CCF/PEEK wrapped yarn will cause the wire feeding difficulty. It is necessary to upgrade the existing 3D printer to be more suitable for printing of wrapped yarn.”

Round 2

Reviewer 1 Report

The  authors answered all my questions in the rebuttal and I don't have any comments or concerns.

Reviewer 2 Report

The paper can be accepted as it is